# Nanoscale deformation mechanics reveal resilience in nacre of *Pinna nobilis* shell

Jiseok Gim [1], Noah Schnitzer[1], Laura M. Otter [2], Yuchi Cui[1], Sébastien Motreuil[3], Frédéric Marin[3], Stephan E. Wolf [4,5], Dorrit E. Jacob [2], Amit Misra[1] & Robert Hovden[1,6]*

The combination of soft nanoscale organic components with inorganic nanograins hierarchically designed by natural organisms results in highly ductile structural materials that can withstand mechanical impact and exhibit high resilience on the macro- and nano-scale. Our investigation of nacre deformation reveals the underlying nanomechanics that govern the structural resilience and absorption of mechanical energy. Using high-resolution scanning/transmission electron microscopy (S/TEM) combined with in situ indentation, we observe nanoscale recovery of heavily deformed nacre that restores its mechanical strength on external stimuli up to 80% of its yield strength. Under compression, nacre undergoes deformation of nanograins and non-destructive locking across organic interfaces such that adjacent inorganic tablets structurally join. The locked tablets respond to strain as a continuous material, yet the organic boundaries between them still restrict crack propagation. Remarkably, the completely locked interface recovers its original morphology without any noticeable deformation after compressive contact stresses as large as 1.2 GPa.

[1] Department of Materials Science & Engineering, University of Michigan, Ann Arbor, MI, USA. [2] Department of Earth and Planetary Sciences, Macquarie University, Sydney, NSW, Australia. [3] Laboratoire Biogéosciences, Université de Bourgogne Franche-Comté (UBFC), Dijon, France. [4] Department of Materials Science & Engineering, Friedrich-Alexander-University Erlangen-Nürnberg (FAU), Erlangen, Germany. [5] Interdisciplinary Center for Functional Particle Systems (FPS), Friedrich-Alexander University Erlangen-Nürnberg (FAU), Erlangen, Germany. [6] Applied Physics Program, University of Michigan, Ann Arbor, MI, USA. *email: hovden@umich.edu

The inherent tradeoff between strength and toughness inspires new design approaches to structural materials with high damage tolerance. While plastic deformation degrades materials' strength and performance lifetime, it is the key attribute for toughness and resistance to fracture. Thus overcoming the tradeoff between toughness, strength, and resilience remains a fundamental design challenge for structural materials[1]. Optimizing mechanical properties for predictable and non-catastrophic failure motivates novel design of modern high-performance structural materials[2,3]. Nature has optimized high-performance materials with unrivaled strength, toughness, and resilience using three-dimensional (3D) hierarchical architectures that traverse the atomic, nano-, micro-, to macro-scale with precision that human technology is yet to achieve[4].

Among the diverse set of structural biominerals—such as bone[5], enamel[6], and various biosilica[7]—to be mimicked for designing new synthetic structural materials, nacre is the prototypical supermaterial[1,5,8-10]. After crack initiation, bulk nacre shows a 40-fold higher fracture toughness than the monolithic/single crystal calcium carbonate from which it is constructed[5,11-14]. Thus a central focus has been placed on understanding the principle mechanisms of nacre's excellent mechanical properties to inspire new designs of next-generation high-performance structural materials[5,12-22]. However, nacre's ability to undergo limited deformation and dissipate critical stresses before fracture has not yet been quantified nor correlated with nanomechanical processes.

Nacre is constructed from layered interdigitated polygonal (or pseudo-hexagonal) aragonite ($CaCO_3$) platelets (0.5–1 μm thick and 10–20 μm wide), bonded by a thin (~5–30 nm thick) layer of organic material (the interlamellar membrane). Nacre platelets are either arranged into a brick-and-mortar-like architecture in the sheet nacre of bivalves or are stacked vertically as columnar nacre in gastropods[23-26]. A natural composite material, nacre is reported to consist of roughly 95–98 wt. % aragonite and ~2–5 wt. % biopolymers[16,19,20,26-30]. Our measurements herein confirm a 3.4 ± 1.0 wt. % organic fraction for *Pinna nobilis* (Mollusca, Bivalvia). The organic fraction of nacre consists of organic interlamellar membranes[31] and intracrystalline organics embedded in the mineral tablets[31-33] of ~5–20 nm. Nacre tablets have a textured surface roughness and internal substructure that are both derived from space filling nano-granules[25,34-37]. The surface contains nano-asperities suspected to play an important role in the prevention of tablet sliding[38]. Surface asperities between opposing nacre tablets occasionally form narrow (20–50 nm) intrinsic mineral bridges[33] without external stress (e.g., Supplementary Fig. 1) connecting across the interlamellar membrane, while wider (150–200 nm) major intrinsic mineral bridges are thought to be involved in the initial formation of new nacre tablets[26,31].

In nacre, Wegst et al.[9] have suggested that crack bridging and the resulting "pull-out" of mineral bricks is associated with controlled, yet limited, sliding of the aragonite layers over each other and is aided by visco-plastic energy dissipation in the organic layer. Li et al. observed the plastic deformation of aragonite surfaces under tensile load at the nanometer level using atomic force microscopy[18,21]. However, additional mechanisms for strengthening and toughening have been proposed: resistance from the lamellae nanoroughness[19], the organic layer acting as a viscoelastic glue[7,16,20,22], the presence of (pre-existing) mineral bridges[13,17,30], and platelet interlocking at the microscopic level[22]. Direct observation is required to disambiguate the mechanism of nanomechanical deformation of nacre; however, most knowledge of the biomineral toughening process is assembled from microscale tribology[8,12,15,16], tensile[5,12,15,19,20,22,39], or compression[14,19,40] testing on bulk specimens. Understanding nanomechanical

responses across the 3D hierarchical architectures is critical to understanding how the individual nacre components work together to create properties greater than the sum of their parts (i.e., far exceeding the rule of mixtures[9]).

Our investigation of toughening strategies in nacre reveals nanomechanical deformation of organic interfaces, nanocrystallites, and organic inclusions as key to the increased damage-tolerance of nacre. High-resolution scanning/transmission electron microscopy (S/TEM) combined with in situ nanoindentation[41,42] has been adapted to biomineral systems to allow sub-nanometer resolution imaging of the nanomechanical deformation processes and provide precise assessment of when and where fracture occurs. We show that during compressive indentation nacre undergoes non-destructive locking where inorganic tablets come into contact across organic interfaces. Remarkably, the completely locked interface recovers its original morphology without any deformation after releasing compression and retains its full mechanical strength. During compression, the aragonite grains and organic inclusions reversibly rotate and deform indicating nanoscale resilience of the nacre tablets. Prior to tablet locking, strain attenuates up to 80% between the decoupled tablets. However, by 3% engineering strain of the first tablet, the tablets have locked to redistribute stress continuously across the organic interface and the strain attenuation decreases.

When fracture occurs, we show the organic components restrict crack propagation both within and between tablets, sustaining the overall macroscale architecture through multiple fractures to allow further structural loading. This allows nacre to absorb significantly greater mechanical energy than monolithic aragonite. We report that nacre absorbs roughly 1–3 times more mechanical energy than geological (i.e., non-biogenic) monolithic aragonite before fracture results in structural failure under nanoindentation. This approach provides an energy dissipation measurement that is not derived from a crack-propagating force. In addition, we show that the yield strength measured under nanoscale compression along the *c*-axis (growth direction) of a single tablet can reach values three times higher (e.g., ~1.1–1.6 GPa) than previously reported for bulk nacre measured with microindentation[1,5,13-16,27,38,40,43].

## Results

**Nanoscale deformation and toughening processes.** We observe non-linear elastic nanoscale deformation and toughening processes in nacre under compression using nanoindentation with 0.04–0.2 μm² contact areas approximately normal to the growth direction (*c*-axis). This surface normal is nacre's strongest direction[12,13,38], although the monolithic aragonite from which it is comprised is stiffer along the planar direction[44]. Electron transparent cross-sections from a mature *P. nobilis* specimen (Fig. 1a) provided the structural stability required for indentation while allowing sub-nanometer resolution imaging (see "Methods"). S/TEM observation revealed a range of strengthening and toughening processes enabled by nacre's hierarchical structure: (i) tablet interlocking, (ii) strain damping, (iii) crack blunting, and (iv) intracrystalline deformation and rotation of nanograins and organics. Despite comprising only a few weight percent (i.e., ~2–5 wt. %[16,19,20,26-30]) of the entire nacre architecture, the organic components of nacre provide a range of functions that absorb the energy of applied loads while remaining highly recoverable even after initial fracture. The ratio of high-angular annular dark-field (HAADF) STEM intensity estimates the total organic volume fraction in *P. nobilis* nacre to be 7.1 ± 2.2 vol. % (3.4 ± 1.0 wt. %) comprised of 2.5 ± 0.3 vol. % (1.2 ± 0.1 wt. %) interlamellar and 4.6 ± 1.9 vol. % (2.2 ± 0.9 wt. %) intracrystalline material (see Supplementary Fig. 2).

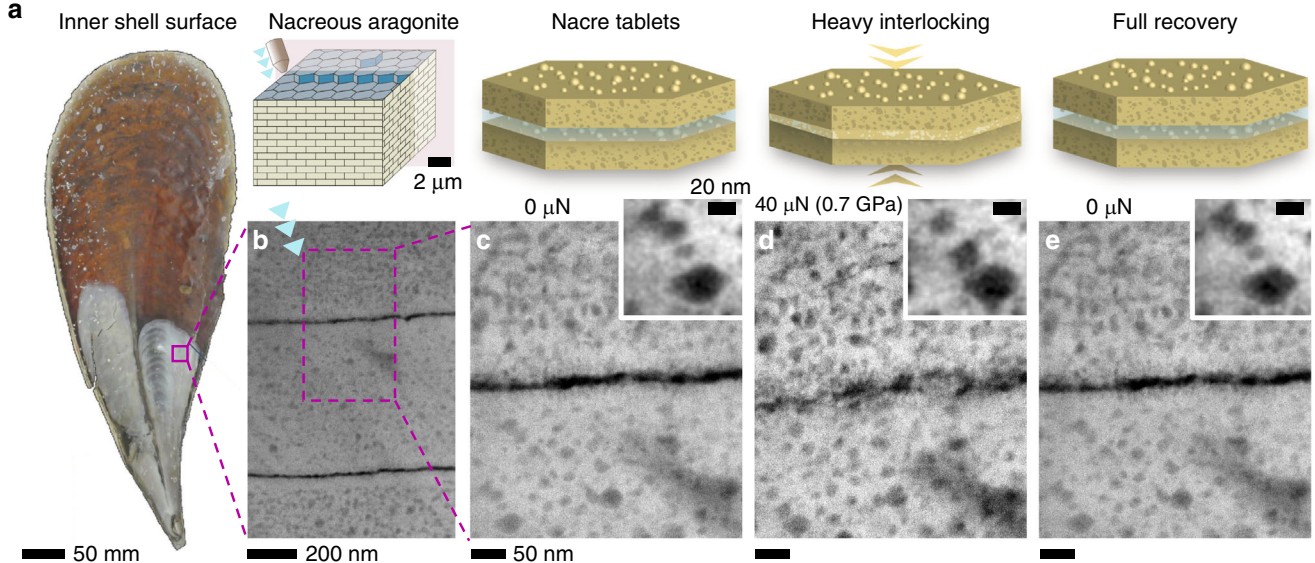

**Fig. 1** Highly deformed and recovered nacre. **a** Schematic of the inner shell surface of the bivalve mollusk *P. nobilis*, with the investigated area marked by a purple square. **b** HAADF STEM overview image of cross-sectional interface of nacre tablets before compression. **c** High-resolution STEM image of two tablets and their organic interface before compression. **d** Tablets heavily interlocked under 40 μN compressive load. **e** After indenter is retracted, tablets and organic interface have fully recovered their initial morphology. Insets show the movement of organic inclusions due to the deformation of the tablet and their complete recovery after removing the compressive load

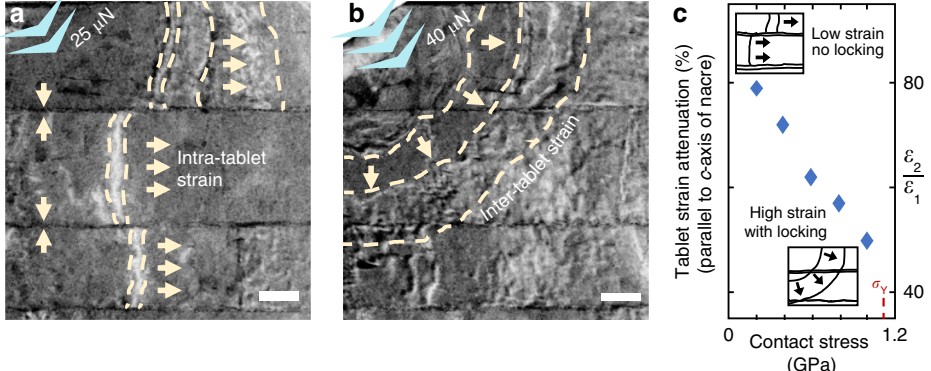

**Fig. 2** Strain propagation confined by organic interfaces. **a**, **b** Bright-field TEM (with contrast inverted) on the cross-sectional nacreous region under low and high compressive contact stresses. Under low compressive stress, intra-tablet strain contours are generated, and strain propagates independently along each tablet. As the compressive stress is increased, nacre tablets interlock and larger inter-tablet strain contours propagate diagonally between tablets. **c** Tablet strain attenuation along the axis of the indentation source. The linear strain dissipation behavior indicates that the deformability of nacre is weakened as the applied stress is increased. Scale bar 200 nm

**Highly recoverable nacre tablet locking**. Nacre's nanoscale organic boundaries and inclusions allow heavily deformed nacre to fully recover its original morphology on the nanoscale (Fig. 1, Supplementary Figs. 3 and 4). Under large compressive loads (e.g., 0.7 GPa in Fig. 1d), opposing nacre tablets interlock across the mineral–organic interface to form temporary inorganic connections through the joining of asperities. Further, the entire tablet volume compresses resulting in small but discernable deformation of organic inclusions (Fig. 1). After releasing the load, the mineral connections at the deformed organic interface and the intratablet nanostructure perfectly recover their initial morphology without any sustained deformation (Fig. 1e, Supplementary Fig. 3a–d). This remarkable recovery after tablet locking was reproduced and observed across all areas of interest (Supplementary Figs. 3 and 4).

Nacre shows mechanical response regimes of high and low compression visible in the strain contours measured during in situ TEM indentation (Fig. 2). Low compressive loads applied along the growth direction generate strain contours, which propagate transversely within each tablet (Fig. 2a). Shearing of the interlamellar membranes prevent propagation longitudinally to neighboring tablets. At higher loads, tablets couple, coming into direct contact with one another and allowing strain contours to spread across tablets radially from the location of indentation (Fig. 2b). Strain along the *c*-axis is highest directly below the tip loading and tablet compression (tablet engineering strain) is measurable using interlamellar demarcation (Supplementary Fig. 6). By ~3% engineering strain in the first tablet, the contours redistribute continuously, and by ~6% engineering strain, locking is strikingly visible between tablets. Initially, the engineering

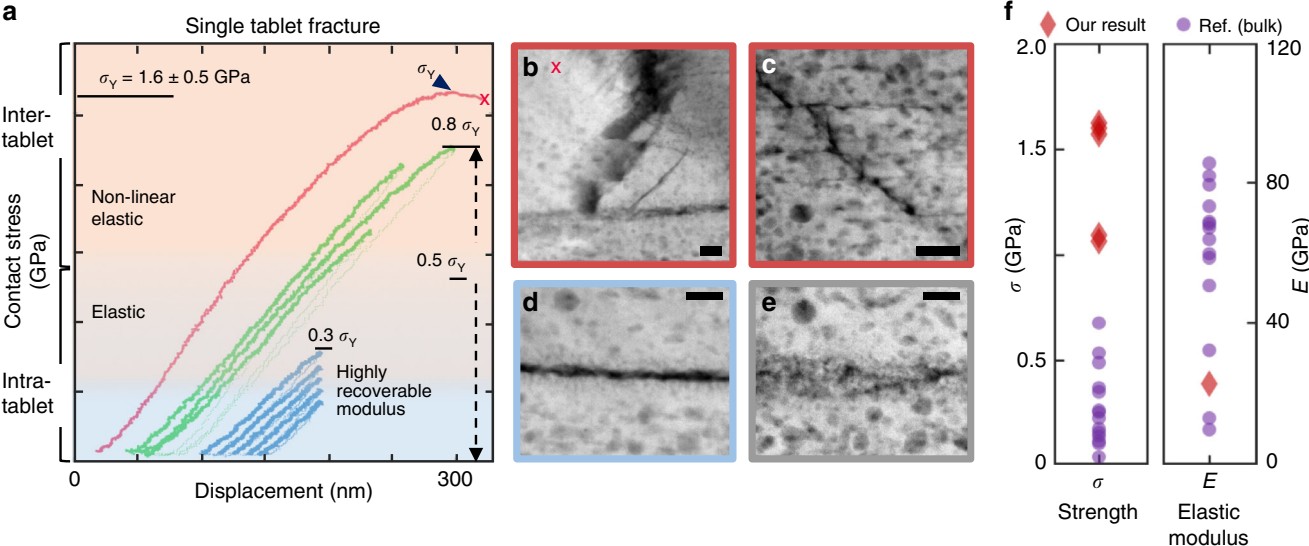

**Fig. 3** Recoverable mechanical strength of nacre and crack blunting within and between tablets. **a** Nine consecutive in situ TEM compression tests on the same nacreous tablet. Three different colors correspond to different contact areas during the series of the compressions. Stage drift caused changes to contact area between indentations. **b** ADF STEM images after the series of the indentation tests showing a crack blunted by an organic boundary. **c** ADF STEM image shows crack formed within tablet and blunted by an organic inclusion. **d, e** ADF STEM images of nacre tablet compressed by 47 µN (55% of $\sigma_{Yield}$), corresponding to the non-linear elastic regime; structure remains fully recoverable—after deformation, nacre still preserves both its initial strength and structure. f Strength and elastic modulus of nacre from contact stress in nanoindentation on the thin cross-sectional specimen in this study and various types of testing—microscale tribology, tensile, compression, and bending—on bulk specimens in previous reports. Scale bar 50 nm

strain of the first to second tablet along the axis of indentation decreases by >~80% when measured using a 0.1-µm² contact area. This measurement is only one component of an inhomogeneous strain field that, on average, dissipates away from the point of compression. As greater contact stress is applied, the tablets increasingly lock farther away from the tip and the strain attenuation linearly decreases—the deformability is reduced as the nacre behaves more like a monolithic material (Fig. 2c). This entire process occurs with elastic processes that are fully repeatable. We note that tablets also exhibit a limited amount of locking for indentation parallel to the tablet plane (Supplementary Fig. 5). This occurred near the indentation tip where stress is high, and the Poisson effect pushes the compressed tablet against its neighbors. Further away, unlocked tablets accommodate shear deformability at their interface and strain contours are discontinuous. Indentation parallel to the tablet plane was less resilient and typically resulted in unrecoverable brittle fracture (Supplementary Fig. 5).

**Preservation of mechanical strength.** During consecutive indentation tests, highly deformed nacre fully recovers under external loads up to ~80% of its yield contact stress. This can be seen in Fig. 3a, where the elastic modulus remains unchanged during eight consecutive compressions (blue and red). As shown in the specimen of Fig. 3d, e, beyond ~0.8 GPa, nacre begins nonlinear elastic deformation—yield is visible from the decreasing slope of the contact stress–displacement curve. However, unlike traditional plastic deformation, the initial structure is preserved after unloading. Full recovery was even observed in highly deformed nacre (e.g., >~0.8–1.1 GPa) prior to crack formation (Fig. 1, Supplementary Figs. 3 and 4). This preservation of mechanical strength under repeated loading/unloading cycles reflects a non-linear elastic deformation process featuring nano-mechanical resilience not present in traditional bulk materials, attributable at least in part to tablet interlocking. The rotation and deformation of organic inclusions and nano-granularity has also

been predicted as another mechanism for viscoelasticity[21]. Although structurally recoverable locking of tablets is key to nacre's resilience, in the reported nanoindentation experiments, performed under dry conditions, the absorbed energy appears to primarily occur within the resilient deformation of nanograined tablets that constitute a significant volume fraction (~97%). This process is confirmed in our bright-field TEM data, where individual aragonite nanograins change contrast as nanograins reorient and organic inclusions slightly reshape their volume (Supplementary Fig. 7, Supplementary Movie 1).

The deformation of these nanometer-scale organic inclusions with compression of the material accommodates the load while avoiding irrevocable damage to the inorganic matrix (Fig. 1c–e, also shown in insets of Supplementary Figs. 3, 4 and 7). Here nacre's response shows non-linear elastic deformation distinct from that expected in analogous nanocrystalline metals. Unlike nano- or micro-grained metals, which strengthen through reduced mobility of dislocations at grain boundaries[45,46], nacre's proteinaceous organic components contain flexible molecular bonds that elastically accommodate strain and rotation of nanograins and restoratively return the system to the original state when an external stress is released. This process occurs without the introduction of dislocation pile-up and plastic deformation. Energy absorption during protein stretching/unfolding and subsequent energy release upon refolding of the elastomeric molecules provides high resilience in nacre and similar to that found in bone[47]. In contrast, nanocrystalline or nanotwinned metals have lower resilience since they exhibit plasticity via dislocations.

At failure, the organic components in nacre impede crack propagation both within and between tablets (Figs. 3b, c and 4c, Supplementary Fig. 8). The smaller organic inclusions embedded within the inorganic matrix hinder crack propagation within the tablet and were observed to blunt and deflect cracks (Fig. 3c). The interlamellar membrane likewise hampers propagation between tablets, where cracks often terminate or jump at the interface

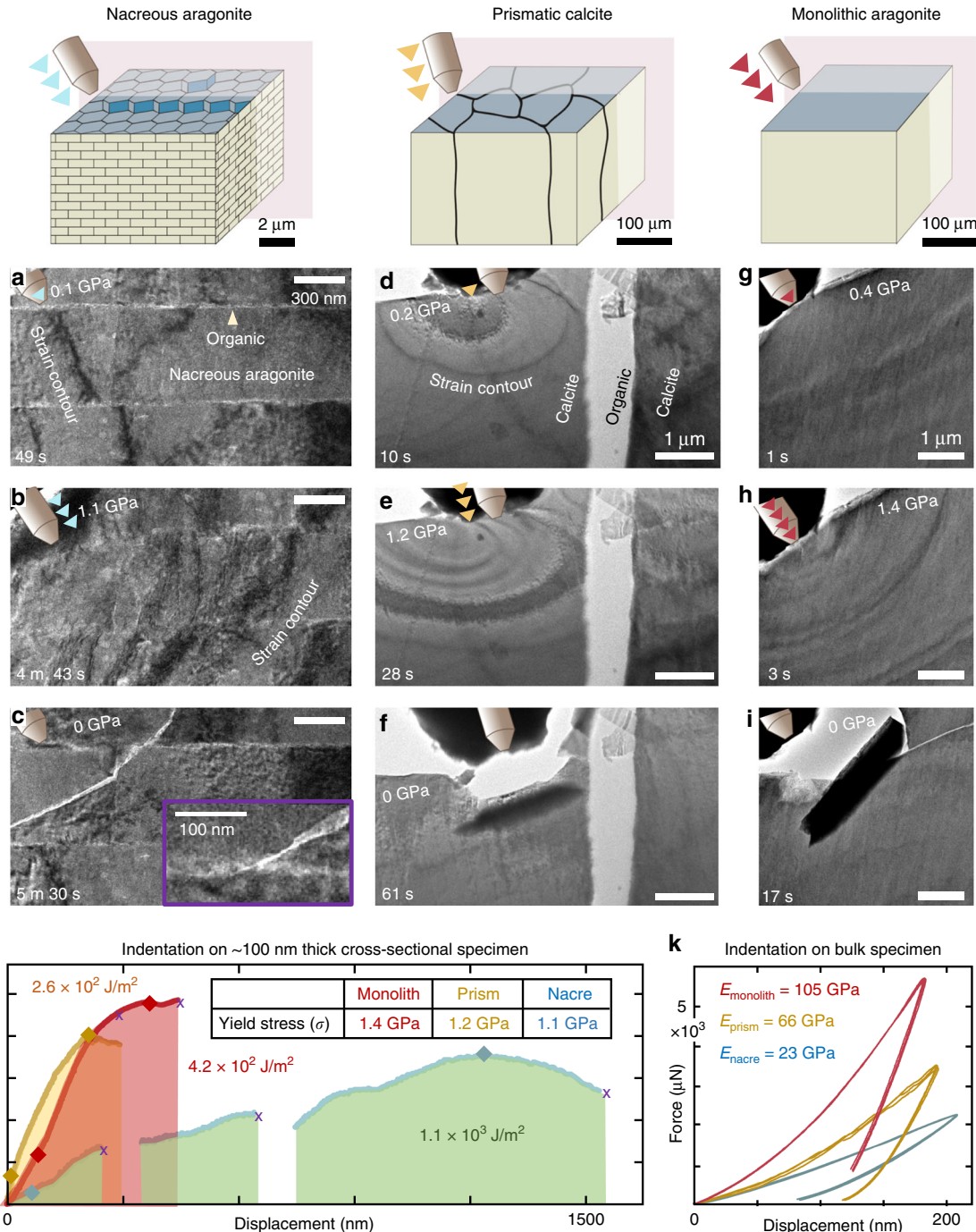

**Fig. 4** Toughening processes of nacre, prismatic calcite, and monolithic aragonite. **a–c** Bright-field TEM images of the cross-sectional nacreous region during in situ TEM indentation. The nacreous tablets made contact on the side of the tip (tip diameter: ~100 nm). Inset in **c** shows crack blunting at the organic interface. **d–f** Bright-field TEM images of the cross-sectional prismatic calcite region during indentation. **g–i** Bright-field TEM images of non-biogenic, monolithic aragonite during indentation. **j** Correlative compressive contact stress vs. displacement curve showing mechanical response of the nacreous, prismatic, and monolithic region. Stress herein is engineering stress calculated by dividing load by cross-sectional area contacted with tip. Total energy dissipation values (area under contact stress–displacement curves) marked. **k** Triboindentation on bulk specimens of nacre, prismatic calcite, and monolithic aragonite. Videos provided as Supplementary Material. (See Supplementary Movies 2, 3, and 4)

(Fig. 3b). After each fracture event, the overall macroscale architecture is preserved and maintains its mechanical properties (Fig. 4a–c, Supplementary Fig. 9, Supplementary Movie 2). This extends the damage tolerance of the superstructure beyond a single fracture. In fracture mechanics[48], the ability to resist fracture is quantified by a fracture toughness when a crack is present. In this complex material, local stress states can lead to a variety of

mechanisms responsible for the fracture process zone. Here cracks can be under mixed mode loading conditions, which in general can lead to differences in the energy required for crack extension and make quantification of fracture toughness by nanoindentation challenging. In bulk specimens loaded in mode I, a fracture toughness of 10 MPa·m$^{1/2}$ has been reported for nacre, 40-fold larger than that of single crystal aragonite, ~0.25 MPa·m$^{1/2}$ [14,39].

**Damage-tolerance of nacre's architecture**. On a system level, nacre can sustain several fractures before total failure due to its hierarchical soft–hard matter design. This means the yield stress of nacre is not typically defined by crack initiation. In contrast, prismatic calcite and monolithic aragonite exhibit limited deformation before the yield stress is followed by catastrophic failure or crack runaway (Fig. 4d–i, Supplementary Figs. 10 and 11, Supplementary Movies 3 and 4) from cone cracking at indentation. Monolithic aragonite responds to strain with stress contours radiating from the point of contact. Prismatic calcite from the *P. nobilis* mollusk behaved similar to monolithic aragonite; however, indentation near an organic interface showed significant attenuation into an adjacent prism (Fig. 4e).

When compared to monolithic calcite materials, we clearly see nacre's interlamellar membranes reshape compressive strain fields. Both biogenic calcite from the prismatic layer of *P. nobilis* and geological monolithic aragonite were noticeably stiffer than nacre (Fig. 4k) and typically reached higher yield stresses than nacre (Fig. 4j). However, nacre's inorganic–organic architecture reliably absorbed 1–3 times more mechanical energy than prismatic calcite and monolithic aragonite before total failure. Integrating the applied stress over the displaced volume of the indenter contact area provides an upper bound on nacre's energy dissipation of $1.1 \times 10^3$ $J/m^2$. Here nanoindentation provides us with true estimates of the energy required to cause fracture(s) that lead to structural degradation. A typical contact stress–displacement curve for nacre often included several intermediate failures, where cracking was halted, nanoscale morphology of nanograins and organic inclusions was preserved, and nacre could undergo further loading without a noticeable change of structure in its mechanical response. Notably in calcite and monolithic aragonite, crack runaway occasionally allowed noticeable energy absorption—however, this occurred after the maximum yield stress and resulted in the unrecoverable structural failure typically found in brittle materials.

In situ nanoindentation enables mechanical behavior to be measured at the single tablet level, allowing the contributions of the toughening and resilience mechanisms across length scales to be assessed. For instance, while the elastic modulus of nacre and calcite from *P. nobilis* were comparable to previous reports on bulk specimens[14–18,27,38,49–54] (Fig. 3f, Supplementary Fig. 12), the measured strength of nanoindented nacre reached values as high as $1.6 \pm 0.2$ GPa, roughly three times larger than the literature reports for bulk nacre in hydrated and dehydrated specimens[1,5,13–16,27,38,40,43]. Dehydrated nacre has been shown to have a greater strength and elastic modulus but lower toughness than hydrated nacre due to the plasticizing of the organic matrix by water[12]. Here the *P. nobilis* specimen was sacrificed and dehydrated. In native conditions, the performance of nacre should be even better; we underestimate the recoverability of nacre under conditions of low pressure and low hydration and overestimate its tendency to fracture. Typically, nanoindentation in the thin cross-sectional specimens of the nacre and calcite from *P. nobilis* and geological monolithic aragonite also resulted in a yield strength (e.g., $1.1 \pm 0.1$ GPa) larger than previously reported bulk values (Fig. 3f, Supplementary Fig. 13). The high strength may be attributed to the finite size of the indentation tip and nanoscale size effects of the mechanical response. As previously observed in several materials—including gold nanowires[55], polycrystalline thin films[56], and multiwalled carbon nanotubes[57]—the size effects on mechanical properties of nanostructured materials deviate from bulk and necessitate the use of in situ nanomechanical testing[58,59].

## Discussion

The present in situ S/TEM nanoindentation study illuminates nacre's distinct non-linear elastic deformation processes that provide high resilience. We see how large forces can drive nacre into locked states that allow the material to distribute strain across tablets and recoverably absorb energy through inorganic and organic compression, nanograin reorientation, and the deformation of organic inclusions. After the load is removed, locked nacre completely recovers both its original morphology and mechanical strength. Even after fracture, failure is mitigated through barriers to crack propagation that preserve the macroscale architecture and allow nacre to retain its mechanical properties and further sustain impact. The material's structure and deformation mechanisms allow it to absorb more mechanical energy than geological monolithic aragonite and biogenic prismatic calcite. Using in situ S/TEM nanoindentation, the mechanical properties of the material were tested down to the individual tablets where yield ~3 times stronger than bulk measurements were observed.

This approach enables investigation of the wider range of evolutionary-optimized biominerals to reveal advantages underlying their nanomechanical design. The study of deformation and fracture under nanoindentation is a subset of the broader fracture phenomena in nacre and other biological materials, which may reveal additional nanomechanical responses to external forces such as tensile strain or shear. The observed mechanisms reported in this work may guide theoretical models of deformation behavior, and the demonstration of in situ S/TEM nanoindentation of nacre opens the possibility of other in situ S/TEM such as bending, tensile, etc., across a wider range of biological and bioinspired composites. For nacre under compression, the rich multiscale resilient deformation processes and interlamellar locking inspires new synthetic routes to complex structural materials.

## Methods

**Specimen**. Specimens of the protected Mediterranean *P. nobilis* (*Pinnidae*, Linnaeus 1758) bivalve species were live collected in the bay of Villefranche-sur-Mer, Département Alpes-Maritimes, France. All necessary permits were acquired from DDTM (Direction Départementale des Territoires et de la Mer) of Alpes-Maritimes department. *P. nobilis* is strongly protected by a European Directive (92/43/CEE). Specimens of the geological monolithic aragonite were mined in Sefrou, Morocco.

**Sample preparation**. After the bivalves were sacrificed, small shell sections were cut from the whole shell measuring 60 cm shell height[34] using a diamond wire saw. To avoid beam damage and amorphization from ion beam milling, cross-sections for S/TEM were prepared by mechanical wedge polishing[34]. This technique provided large-area, electron-transparent specimens with structural stability critical for nanotribology. Nacre samples had thicknesses of $124 \pm 3$ nm (Fig. 4a–c) and $98 \pm 2$ nm (Fig. 3a), prismatic calcite had a thickness of $102 \pm 6$ nm (Fig. 4d–f), and monolithic aragonite had a thickness of $169 \pm 1$ nm (Fig. 4g–i).

**Electron microscopy**. Real-time observation of the compressive nano-deformation was performed using S/TEM. Column pressure in the TEM column of the specimen was $\sim 1 \times 10^{-7}$ torr. Bright-field TEM with 60 μm (for nacre) and 120 μm (for biogenic prismatic calcite) apertures provided contrast of strain contours and performed on a 200 keV JEOL 2010F and Gatan OneView camera enabling frame rates of up to 200 frames per second. Images were captured at 50 frames (2048 × 2048 pixels) per second for nacre and at 12.5 frames (2048 × 2048 pixels) per second for biogenic prismatic calcite and monolithic aragonite. STEM was performed using a JEOL 3100R05 microscope with Cs aberration corrected STEM (300 keV, 22 mrad) and cold field emission gun. A HAADF detector with 120–150 mm camera lengths and a detector angle from 59–74 (inner) to 354–443 mrad (outer) were used to produce Z-contrast images where grayscale intensity is sensitive to the atomic number in the specimen's matrix.

No change was observed in mechanical behavior measurements with beam exposure: whether the beam was blanked or the microscope was operated in low-dose STEM or TEM mode. Low-dose methods, beam shuttering, and examination of regions exposed to the beam were used to separate electron beam irradiation from intrinsic phenomena. For STEM measurements, with a typical field of view of 500 nm the electron dose was typically $\sim 150$ $e^-/Å^2$ and dose rates around $\sim 4$ $e^-/Å^2/s$; the material was structurally preserved during imaging. However, for the same imaging conditions at higher magnifications (e.g., 30 nm field of view) the radiation dose increases to $\sim 2 \times 10^5$ $e^-/Å^2$ and dose rate to $\sim 10^3$ $e^-/Å^2/s$, which causes the material to show electron irradiation damage localized to the small field of view. Thus larger fields of view are preferred to minimize dose and provide a large area of observation where fracture may nucleate. Atomic-resolution STEM requires a small field of view, on-axis region of interest, and a static specimen, which was not

achievable during in situ nanoindentation. This limits atomic imaging during nanoindentation despite the well-aligned instrument's probe-limited resolution of ~1 Å. For TEM imaging, dose was minimized through use of a heavily diverged beam and a high-efficiency camera (DQE of 0.3) with single electron sensitivity and high-readout speed (up to 300 fps).

Low-loss electron energy loss spectra (EELSs) were acquired at 300 kV with a Gatan Quantum Energy Filter, with 1.5 eV per channel to determine thickness of the specimen. The convergence semi-angle was 22 mrad and the collection semi-angle was approximately 40 mrad. Linear combination of power laws with local background averaging was applied to analyze the spectrum image using the Cornell Spectrum Imager[60]. Thickness of the specimen is determined from plural scattering in EELS, which is defined by $I_0 = I_{ZLP} \cdot e^{-\frac{t}{\lambda}}$, where $I_0$ is total plural scattering in electron-loss spectrum, $I_{ZLP}$ is zero loss peak in the spectrum, $t$ is the thickness of the nacre, and $\lambda$ is the wavelength of the incident electron beam, as described in Egerton[61]. Relative organic concentration in nacre tablet was formulated by ratio of high-angle elastic electron scattering intensity, which is defined by $I_{HAADF} = t \cdot \sum Z_x^{\gamma} \cdot \rho$, where $I_{HAADF}$ is the HAADF intensity, $t$ is the thickness of the nacre, $x$ is a certain element in $CaCO_3$ or organic molecule, $\gamma$ is elastic scattering cross-section ranging from 1.4 to 1.7, and $\rho$ is the $CaCO_3$ or organic molecular density (Supplementary Fig. 2).

**In situ nanoindentation**. Nanoindentation experiments were conducted in the TEM column (~25 °C, $10^{-6}$ Torr) using a Hysitron PI-85 PicoIndentor. Load-controlled nanoindentation was performed using cube corner (tip radius = ~0.1 μm, half-angle = 35.26°, included angle = 90°) and conospherical (tip radius = ~1 μm, semi-angle = 60°) diamond probe tips. Maximum loads varied from 10 to 400 μN. A piezoelectric actuator controlled the specimen position in all three dimensions. During indentation, the indenter was advanced at a rate of 5 nm/s for nacreous aragonite and prismatic calcite and at a rate of 60 nm/s for monolithic aragonite. Force–displacement information and movies were recorded during indentation, and still TEM micrographs were collected between tests. The electrostatic force constant of the transducer was calibrated such that the root-mean-square error fell <~$10^{-5}$ μN/V² using Z-axis calibration, which results in the measurement error in force and displacement within ±5%. The top surface of the tip was aligned vertically to the cross-sectional nacre specimen to achieve uniaxial compression without shear or bending. For all the samples, contact stress is calculated by dividing the measured load by cross-sectional area of the specimen in contact with the indenter tip. This contact area is estimated by multiplying the length of the contact region measured in real time with S/TEM images and the specimen cross-sectional thickness measured by the ratio of zero loss/total low-loss EELS. Total error of contact stress is calculated by a quadrature of the errors from the contact length measured by human vision (±10%), the specimen thickness estimated by EELS image (±11%), and the load reported by the nanoindentation software (±5%). The contact area changes through subsequent indentations (Fig. 3) due to stage drift (typically 20–60 nm). Toughness (J/m²) is defined as the absorbed mechanical energy, which can be bounded by integrating the stress–displacement curve to find the energy absorbed per unit area (Fig. 4j). The tablet engineering strain along the c-axis is defined as the ratio of the reduction of the tablet width (that is, compressive deformation directly under the region of loading) to its initial width (Supplementary Fig. 6). Strain attenuation is defined as the ratio of the measured tablet engineering strain between the first and the second tablet from the indenter tip contact location (Supplementary Fig. 6).

**Triboindentation**. Triboindentation experiments were carried out on bulk biological aragonite, calcite, and geological aragonite samples (5 × 5 mm² area, 3 mm thick) with polished surfaces to determine the elastic modulus of the materials using a Hysitron TI-950 Triboindenter. During indentation, the indenter was advanced at a rate of 20 nm/s for nacre, prismatic calcite, and monolithic aragonite with a Berkovich tip (i.e., three-sided pyramidal diamond tip). The probe area function was calibrated for the Berkovich tip, particularly in the low-depth ranges using a standard quartz sample before determining the mechanical properties accurately. To validate the tip calibration, a standard Al single crystal was used to confirm that our calibration values match the elastic modulus and hardness (69.6 ± 10% and 9.25 ± 10% GPa) provided by the manufacturer within a standard deviation of 5% using the standard Oliver–Pharr method.

**Reporting summary**. Further information on research design is available in the Nature Research Reporting Summary linked to this article.

## Data availability

The authors declare that the source data underlying the main Figs. 1a–e, 2a–c, 3a–e, and 4a–k are provided as a Source Data file. All other relevant data supporting the findings of this paper are available from the corresponding author upon reasonable request. Supplementary Movie 1: https://doi.org/10.6084/m9.figshare.9869888. Supplementary Movie 2: https://doi.org/10.6084/m9.figshare.9869912. Supplementary Movie 3: https://doi.org/10.6084/m9.figshare.9869915. Supplementary Movie 4: https://doi.org/10.6084/m9.figshare.9869918

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

## Acknowledgements

The authors acknowledge the University of Michigan College of Engineering for financial support and the Michigan Center for Materials Characterization for use of the instruments. S.E.W. acknowledges financial support by an Emmy Noether starting grant issued by the German Research Foundation (DFG, no. WO1712/3-1). We thank Kai Sun for experimental assistance and Anna Son for illustration assistance.

## Author contributions

Sample preparation was developed by R.H. and J.G.; electron microscopy was conducted by J.G., N.S. and Y.C.; data analysis and materials' interpretation was carried out by J.G., N.S., A.M. and R.H.; biological context was provided by L.M.O., S.E.W. and D.E.J.; S.E.W., S.M. and F.M. provided specimens; all authors discussed the results and commented on the manuscript. J.G., N.S., L.M.O. and R.H. wrote the manuscript.

## Competing interests

The authors declare no competing interests.
