## [Peer Review File · Nature Communications]

Reviewers' comments:

Reviewer #1 (Remarks to the Author):

What are the major claims of the paper? Are they novel and will they be of interest to others in the community and the wider field?

(1) Under the subtitle "Nanoscale deformation and toughening processes". The authors wrote that "We observe non-linear elastic nanoscale deformation and toughening processes" and they found that "the organic components of nacre provide a range of functions that absorb the energy of applied loads while remaining highly recoverable even after initial fracture.

The authors did have provided evidences on the "non-linear elastic nanoscale deformation", but evidences for the "toughening processes" are not quantitatively convincing with the stress-displacement relationship for the stress measured in the area under the tip. The finding on "non-linear elastic deformation" is trivial, since nacre is a very complicate mineral-protein composite. Also, evidences for the many "functions of organic components" are not all direct. Maybe more specific descriptions/reasoning or arguments are needed.

(2) Under the subtitle "Highly recoverable nacre tablet locking", the authors found that "Under large compressive loads, opposing nacre tablets interlock across the mineral-organic interface to form temporary inorganic bridges through the joining of asperities".

This conclusion is supported by their imaging results shown in Fig. 1d, with a high resolution image shown in Supplementary Figure 1. It is noticed that, however, there is higher resolution image of the mineral bridges published by Checa, et al. in Journal of structural biology 176, 330-339 (2011).

(3) Under the subtitle "Preservation of mechanical strength" the authors claim that "During consecutive indentation tests, highly deformed nacre fully recovers under external loads up to ~80% of its yield stress.

This conclusion is based on their stress-displacement curves. I am not convinced that the strength and toughness of nacre as a macroscopic structure can be quantified by nano-indentation experiments. How to define the toughness and strength of nacre from the indenter's force-displacement curves is not clear. Nacre is a multi-scale structure. Local yielding and cracking may not lead to ultimate failure of the structure.

The authors also wrote that they provided "subnanometer resolution imaging of the nanomechanical deformation processes and provide precise assessment of when and where fracture occurs". The resolution does not seem to be as high as that in the work by Checa, et al. in Journal of structural biology 176, 330-339 (2011).

Is the work convincing, and if not, what further evidence would be required to strengthen the conclusions? do you feel that the paper will influence thinking in the field?

This work used new testing and measurement techniques. The authors also provided many images and records of the testing results. These results are useful and interesting, but a very exciting new finding is not there. I am not sure whether there is a page limit. What is new and what is significant are not articulated convincingly. The manuscript may be improved if the authors can improve their arguments and articulate their novel discoveries. Maybe the results section and the discussion section are too short.

Reviewer #2 (Remarks to the Author):

This paper reports on the in situ TEM deformation and fracture of nacre under nanoindentation. Challenging in situ TEM experiments were performed, which enabled high resolution direct visualization of the processes for the first time. Quantification of force and displacement facilitated quantification of energy dissipation. As such, the manuscript reports new information of interest to the readership of Nature Communications.

General conceptual comments:

1. The study of deformation and fracture under nanoindentation "is a subset" of the broader fracture phenomena in nacre and other biological materials. This distinction is very important for the reasons later articulated and should be mentioned in the introduction and conclusions for the benefit of readers not fully versed in fracture mechanics. To appreciate this point, consider three-point bending fracture experiments as reported in the literature (e.g., Barthelat and Espinosa, *Experimental Mechanics*, Vol. 47, No 3, 2007). In such configuration, the crack is loaded in mode I and the toughness arises from tablet sliding and development of a millimeter process zone. Interfaces between tablets are loaded primarily in shear. By contrast, in the indentation experiments reported in this work, tablet locking arises from interface compressive stresses, which in general may not be present in a propagating crack. Hence, it is important to keep in mind that local stress states can lead to a variety of mechanisms responsible for the fracture process zone. Cracks can be under mixed mode loading conditions, which in general can lead to differences in the energy required for crack extension. Because of this, defining toughness as the area under the stress-strain curve is in general a poor descriptor of fracture toughness and should be avoided. Consider for instance the stress-strain curve one would obtain if instead of compression, the specimen is subjected to tension along the same orientation. I suggest the authors simply employ the area under the stress-strain curve as an indicator of energy dissipation under compression and avoid its generalization to toughness.

2. In relation to the comparison reported in Fig. 4, it is important to note that such comparison involves not only the presence or absence of interfaces with different morphologies but also differences between tablets, containing inorganic nanograins embedded in an organic matrix, and prismatic/single crystals. This should be highlighted because the main contributor to energy dissipation appears to be the structure of the constituents and not the interfaces. To appreciate this point, consider the analogy of metals with structures of nanograins, micrograins and single crystal. They also would behave very differently. Hence, if bio-inspiration is the motivation for the present study, additional analysis and analogies could be used to infer design rules.

The authors should reflect on these two comments to improve clarity of the manuscript and modify it accordingly.

Other comments:

1. On Page 5: "Damping from" should be replaced by "Shearing of" in the opinion of this reviewer.

2. The authors should comment on the implication of interface shear locking on fracture. The interface shear deformability and strength appear to be highly dependent on the normal component of the traction. If the interface is subjected to mainly shear traction, under what conditions would interface locking take place? Does the experimental data reveal any insights on this?

3. On page 6: A) What is the evidence to attribute interface locking a role on the nonlinear elastic deformation process featuring nanomechanical resilience? In other words, if the tablets were 3-4 times thicker, would the material lose resilience? B) The fracture toughness of nacre was first reported by the Espinosa group in the *Journal of Experimental Mechanics*, Vol. 47, 2007. The authors may wish to use this reference at the end of the second paragraph.

4. The statement that identifying size effects on mechanical properties of nanostructured materials necessitates the use of nanoindentation is not fully accurate. A number of other experimental configurations, e.g., membrane deflection and small-scale tension tests were established to assess size effects in thin films, nanowires, and other nanomaterials (Espinosa et al., *JMPS*, Vol. 52, 2004;

Ramachandranmoorthy et al, ACS Nano, Vol. 9, 2015; Bhowmick et al, MRS Bulletin, Vol. 44, 2019). The authors may wish to revise this.

5. Could the authors discuss the molecular origin for the “distinct nonlinear elastic deformation processes that provide high resilience”? Would a nanograined metal or a heavily twinned micrograined metal exhibit the same resilience? If not, what is different and why it matters?

Reviewer #3 (Remarks to the Author):

In their manuscript submitted to Nature Communications, Gim et al. use in situ TEM nanoindentation measurements on different forms of nacre to provide direct observations of previously postulated deformation mechanisms in these materials. While most of the mechanisms have been studied before (and referenced in their paper), I appreciate the systematic approach to this problem by using different microstructural configurations of nacre and in situ observations. The main conclusions of the paper are in line with previous studies of these systems, but with some direct observations of the interlocking behavior and recovery of the organic biopolymer.

However, the paper doesn't address what I would consider as significant potential artifacts, and the calculations of the mechanical data are difficult to interpret. Specifically- the authors seem to calculate stresses and strains readily in situations that most in the nanomechanics community would hesitate to do and also seem to ignore the most obvious questions about in situ TEM mechanical testing of organic beam-sensitive materials.

Here I list what I consider most important problems with the paper:

- 1) The calculations of strain attenuation (ie- the 3% and the 80% I figure 1) are difficult to interpret. How meaningful they are in terms of real strain during a mechanical test (is the strain homogeneous through the sample? Is there a direction or a gradient? Is this true strain or engineering strain? Etc.). The supplementary figure 5 describes the method, I guess, but there seems to be mixing of elastic and plastic deformation, so is strain attenuation referring to elastic or plastic strain or both? If it refers to plastic strain then how meaningful is it- for example, in compression one can simply continue crushing the sample to increase the 'strain attenuation'. My point is that I am not sure how meaningful this measurement is.
- 2) Furthermore, the calculation of strain leaves some unanswered questions. The indentation stress field is quite complicated, varying from tensile at the side to compressive underneath the indenter (shear in between!). So what do these numbers even refer to? I doubt it is possible to accurately quantify stress in this situation- it is difficult enough when you have indentation onto a flat surface, or linear compression of a column, but in this case it is a punch on the end of a thin wedge (at best-most of the figures make it seem that the sample geometries are difficult to define). In fact, the images even show the varying strain field in the strain contours. This is a major problem with this paper.
- 3) Hydration: in their natural state these materials are hydrated and the mechanical response of the biopolymer dried, polished and tested in a vacuum must be different. I was surprised to see no mention of this in the paper.
- 4) Electron beam: similar to the hydration argument, the electron irradiation undoubtedly changes the mechanical properties of the biopolymer (likely as soon as they are first imaged, before any mechanical tests). Again- the authors don't address this directly.
- 5) The observations of rotation and deformation of the aragonite nanograins and organic inclusions (supplementary figure 6) are not at all clear. What exactly is supposed to be rotating in these images and how do you know they are rotating?

Robert Hovden
Dept. Materials Science
University of Michigan
2700 Hayward Ave
Ann Arbor, MI 48109
Phone: (770) 265-4042

August 18, 2019

**Response to Reviewer 1:**

What are the major claims of the paper? Are they novel and will they be of interest to others in the
community and the wider field?

(1) Under the subtitle “Nanoscale deformation and toughening processes”. The authors wrote that “We
observe non-linear elastic nanoscale deformation and toughening processes” and they found that “the
organic components of nacre provide a range of functions that absorb the energy of applied loads while
remaining highly recoverable even after initial fracture.

The authors did have provided evidences on the “non-linear elastic nanoscale deformation”, but
evidences for the “toughening processes” are not quantitatively convincing with the stress-displacement
relationship for the stress measured in the area under the tip. The finding on “non-linear elastic
deformation” is trivial, since nacre is a very complicate mineral-protein composite. Also, evidences for
the many “functions of organic components” are not all direct. Maybe more specific
descriptions/reasoning or arguments are needed.

Our work shows direct observation of how organic components in nacre dissipate external mechanical
energy at the nanoscale: (i) tablet interlocking, (ii) strain damping, (iii) crack blunting, and (iv)
intracrystalline deformation and rotation of nanograins and organics.

We have added an additional measurement, Supplemental Movie 4, to further show the nanograin and
organic response of nacre during indentation.

We have added a discussion on role of organics within the nacre tablet on page 6 para 3, “*Here, nacre’s*
*response shows non-linear elastic deformation distinct from that expected in analogous nanocrystalline*
*metals. Unlike nano- or micro-grained metals, which strengthen through reduced mobility of*
*dislocations at grain boundaries*^{45,46}, *nacre’s proteinaceous organic components contain flexible*
*molecular bonds that elastically accommodate strain and rotation of nanograins and restoratively*
*return the system to the original state when an external stress is released. This process occurs without*
*the introduction of dislocation pile-up and plastic deformation.*”

(2) Under the subtitle “Highly recoverable nacre tablet locking”, the authors found that “Under large
compressive loads, opposing nacre tablets interlock across the mineral-organic interface to form
temporary inorganic bridges through the joining of asperities”.

This conclusion is supported by their imaging results shown in Fig. 1d, with a high-resolution image
shown in Supplementary Figure 1. It is noticed that, however, there is higher resolution image of the
mineral bridges published by Checa, et al. in Journal of structural biology 176, 330-339 (2011).

We agree our imaging results show formation of temporary inorganic bridges (now called “inorganic
connections” on page 5 para 2). The mineral bridges observed by Checa (existing reference [33] is now
added again on page 3) and our corroborating measurement in Supplementary Figure 1 describes
intrinsic mineral bridges. This is distinct from the extrinsic, temporary mineral bridges we report to
form under external stress. To further clarify we now state on page 3 para 1: “...occasionally form
narrow (20-50 nm) *intrinsic* mineral bridges³³ *without external stress* (e.g. Supplementary Fig. 1)
connecting across the interlamellar membrane, while wider (150-200 nm) major *intrinsic* mineral
...^{26,31}.” and also clarify in Supplementary Fig 1 caption with reference to Checa. The already difficult
requirements for achieving on-axis atomic images of CaCO₃ are significantly higher in a dynamic
experiment in a single region of interest, where the specimen moves and the dose per frame is limited.
Dose requirements are now discussed on page 10, para 3 in response to Reviewer #3, comment 4.

(3) Under the subtitle “Preservation of mechanical strength” the authors claim that “During consecutive
indentation tests, highly deformed nacre fully recovers under external loads up to ~80% of its yield
stress.

This conclusion is based on their stress-displacement curves. I am not convinced that the strength and
toughness of nacre as a macroscopic structure can be quantified by nano-indentation experiments. How
to define the toughness and strength of nacre from the indenter's force-displacement curves is not clear.
Nacre is a multi-scale structure. Local yielding and cracking may not lead to ultimate failure of the
structure.

We have revised the text and figures and carefully clarify how our measurements were made and what
our values represent:

As suggested by Reviewer #2 “energy dissipation” best describes our quantified measurement. Page 8,
para 1, now reads “Integrating the applied stress over the displaced volume of the indenter contact area
provides an upper bound on nacre’s *energy dissipation* of 1.1×10^3 J/m². Here, nanoindentation provides
76 us with true *estimates of the energy* required to cause fracture(s) that lead to structural degradation.”
Similarly, Figure 4 now states “*Total energy dissipation* values...” instead of “Toughening values...”

Page 4 para 2 now states on "...This approach provides an *energy dissipation* measurement ...".

In the conclusion, page 9, para 2 we now state, "*The study of deformation and fracture under*
*nanoindentation is a subset of the broader fracture phenomena in nacre and other biological materials*
*which may reveal additional nanomechanical responses to external forces such as tensile strain or*
*shear.*"

In the discussion, page 7, para 2 we now state, "*In this complex material local stress states can lead to*
*a variety of mechanisms responsible for the fracture process zone. Here, cracks can be under mixed*
*mode loading conditions, which in general can lead to differences in the energy required for crack*
*extension and make quantification of fracture toughness by nanoindentation challenging. In bulk*
*specimens loaded in mode I, a fracture toughness of 10 MPa·m^{1/2} has been reported for nacre, 40-fold*
*larger than that of single crystal aragonite, ~0.25 MPa·m^{1/2} 14,39.*"

"Stress" has been replaced with "contact stress" throughout the text and figures and figure captions (Fig
2c, Fig 3a, Fig 4j, Supp. Fig. 5h) when referring to a measurement.

Page 1 para 1 now states on "... after compressive *contact stresses* ...".

Page 5 para 3 now states on "As greater *contact stress* is applied, ...".

Page 6 para 2 now states on "... its yield *contact stress*. ...".

Contact stress is defined (page 11 para 2): "For all the samples, *contact stress* is calculated through
dividing the measured load by cross-sectional area of the specimen in contact with the indenter tip."

The authors also wrote that they provided "subnanometer resolution imaging of the nanomechanical
deformation processes and provide precise assessment of when and where fracture occurs". The
resolution does not seem to be as high as that in the work by Checa, et al. in Journal of structural biology
176, 330-339 (2011).

Our probe-limited resolution is well below 1nm. The atomic resolution images by Checa and our Supp
Fig. 1 occurred in static specimens that permit a serendipitous search for crystallographically on-axis
regions. On page 10, para 3 we now note, "*Atomic resolution STEM requires a small field of view, on-*
*axis region of interest, and a static specimen which was not achievable during in-situ nanoindentation.*
*This limits atomic imaging during nanoindentation despite the well-aligned instrument's probe limited*
*resolution of ~1Å.*" Reviewer #2 notes in our work, "challenging in situ TEM experiments were
performed". We applaud and appreciate the measurements by Checa X. and reference Checa, et al. in
Journal of structural biology 176, 330-339 (2011) in page 2 and now also on page 3.

Is the work convincing, and if not, what further evidence would be required to strengthen the
conclusions? do you feel that the paper will influence thinking in the field?

This work used new testing and measurement techniques. The authors also provided many images and
records of the testing results. These results are useful and interesting, but a very exciting new finding is
not there. I am not sure whether there is a page limit. What is new and what is significant are not
articulated convincingly. The manuscript may be improved if the authors can improve their arguments
and articulate their novel discoveries. Maybe the results section and the discussion section are too short.

We appreciate the reviewer's value of our new testing and measurement techniques and are grateful for
the improvements and references. As requested, we have expanded our results and discussion.

In the discussion on page 9 para 2, we now state: "*The observed mechanisms reported in this work may*
*guide theoretical models of deformation behavior, and the demonstration of in situ nanoindentation in*
*S/TEM capability for nacre opens the possibility of other in situ S/TEM such as bending, tensile, etc.*
*across a wider range of biological and bio-inspired composites. For nacre under compression, the rich*
*multiscale resilient deformation processes and interlamellar locking inspires new synthetic routes to*
*complex structural materials.*" In the results section we have added additional discussion about nacre's
non-linear elastic deformation processes (as discussed with Reviewer #1, comment 1).

We respectfully disagree with this reviewer's value of our manuscript. For the first time, we directly
observed exceptional recoverability and nanoscale self-toughening of a biomineral and nacre's non-
linear elastic responses through interlocking tablets along with organic crack mitigation. Recent
literature on mechanics of nanocomposites (recent reviews [1,2] and also MRS Bulletin June 2019 [3])
highlight the importance of *in situ* TEM nanomechanics in elucidating deformation mechanisms in
small-scale, nano-laminate and nanostructured crystalline metallic materials. However, such *in situ*
nanoindentation S/TEM capability has not been applied to probe the deformation behavior of
biominerals such as nacre.

"The words of Reviewers #2 and #3 best highlight the importance:

To quote reviewer #2 (bold added for emphasis): "This paper reports on the in situ TEM deformation
and fracture of nacre under nanoindentation. Challenging in situ TEM experiments were performed,
which **enabled high resolution direct visualization of the processes for the first time.**
**Quantification of force and displacement facilitated quantification of energy dissipation.** As such,
the manuscript reports new information of interest to the readership of Nature Communications.

And reviewer #3: Gim et al. use in situ TEM nanoindentation measurements on different forms of nacre
to provide **direct observations of previously postulated deformation mechanisms in these**
**materials.** While most of the mechanisms have been studied before (and referenced in their paper), I
appreciate the **systematic approach** to this problem by using different microstructural configurations
of nacre and in situ observations. The main conclusions of the paper are in line with previous studies of
these systems, but with some **direct observations of the interlocking behavior and recovery of the**
**organic biopolymer.**

1. Pro, J. William; Barthelat, Francois; MRS BULLETIN Volume: 44 Issue: 1 Pages: 46-52 Published: JAN 2019.

2. Buehler, Markus J.; Misra, Amit; MRS BULLETIN Volume: 44 Issue: 1 Pages: 19-24 Published: JAN 2019

3. Minor, Andrew M.; Dehm, Gerhard; Advances in In situ Nanomechanical Testing, MRS BULLETIN Volume: 44 Issue: 6
Published: JUNE 2019

**Response to Reviewer 2:**

This paper reports on the in situ TEM deformation and fracture of nacre under nanoindentation.
Challenging in situ TEM experiments were performed, which enabled high resolution direct
visualization of the processes for the first time. Quantification of force and displacement facilitated

quantification of energy dissipation. As such, the manuscript reports new information of interest to the
readership of Nature Communications.

We thank the reviewer for the positive remarks.

General conceptual comments:

1. The study of deformation and fracture under nanoindentation “is a subset” of the broader fracture
phenomena in nacre and other biological materials. This distinction is very important for the reasons
later articulated and should be mentioned in the introduction and conclusions for the benefit of readers
not fully versed in fracture mechanics. To appreciate this point, consider three-point bending fracture
experiments as reported in the literature (e.g., Barthelat and Espinosa, *Experimental Mechanics*, Vol.
47, No 3, 2007). In such configuration, the crack is loaded in mode I and the toughness arise from tablet
sliding and development of a millimeter process zone. Interfaces between tablets are loaded primarily
in shear. By contrast, in the indentation experiments reported in this work, tablet locking arised from
interface compressive stresses, which in general may not be present in a propagating crack. Hence, it is
important to keep in mind that local stress states can lead to a variety of mechanisms responsible for
the fracture process zone. Cracks can be under mixed mode loading conditions, which in general can
lead to differences in the energy required for crack extension. Because of this, defining toughness as
the area under the stress-strain curve is in general a poor descriptor of fracture toughness an should be
avoided. Consider for instance the stress-strain curve one would obtain if instead of compression, the
specimen is subjected to tension along the same orientation. I suggest the authors simply employ the
area under the stress-strain curve as an indicator of energy dissipation under compression and avoid its
generalization to toughness.

We agree that “energy dissipation” better describes our quantified measurement. Page 8, para 1, now
reads “Integrating the applied stress over the displaced volume of the indenter contact area provides an
upper bound on nacre’s *energy dissipation* of \$1.1 \times 10^3 \text{ J/m}^2\$. Here, nanoindentation provides us with
true *estimates of the energy* required to cause fracture(s) that lead to structural degradation.” Similarly,
Figure 4 now states “*Total energy dissipation* values...” instead of “Toughening values...”

In the introduction page 3, para 3 we now state, “We show that during *compressive indentation*...”

In the discussion, page 7, para 2 we now state, “*In this complex material local stress states can lead to
a variety of mechanisms responsible for the fracture process zone. Here, cracks can be under mixed
mode loading conditions, which in general can lead to differences in the energy required for crack
extension and make quantification of fracture toughness by nanoindentation challenging. In bulk
specimens loaded in mode I, a fracture toughness of \$10 \text{ MPa}\cdot\text{m}^{1/2}\$ has been reported for nacre, 40-fold
larger than that of single crystal aragonite, \$\sim 0.25 \text{ MPa}\cdot\text{m}^{1/2}\$ ^{14,39}.”*

In the conclusion, page 9 para 2 we have adapted the reviewers well-written words to now state, “*The
study of deformation and fracture under nanoindentation is a subset of the broader fracture phenomena
in nacre and other biological materials which may reveal additional nanomechanical responses to
external forces such as tensile strain or shear. For nacre under compression, the rich multiscale resilient
deformation processes and interlamellar locking inspires new synthetic routes to complex structural
materials.*”

2. In relation to the comparison reported in Fig. 4, it is important to note that such comparison involves
not only the presence or absence of interfaces with different morphologies but also differences between
tablets, containing inorganic nanograins embedded in an organic matrix, and prismatic/single crystals.
This should be highlighted because the main contributor to energy dissipation appears to be the structure
of the constituents and not the interfaces. To appreciate this point, consider the analogy of metals with
structures of nanograins, micrograins and single crystal. They also would behave very differently.
Hence, if bio-inspiration is the motivation for the present study, additional analysis and analogies could
be used to infer design rules. The authors should reflect on these two comments to improve clarity of
the manuscript and modify it accordingly.

We now highlight the energy dissipation occurs primarily within the structure of the constituencies and
not just at the interfaces on page 6, para 2, *“Although structurally recoverable locking of tablets is key
to nacre’s resilience, the absorbed energy appears to primarily occur within the resilient deformation
of nanograined tablets that constitute a significant volume fraction (~97%).”*

On page 6, para 2 we now state, *“Here, nacre’s response shows non-linear elastic deformation distinct
from that expected in analogous nanocrystalline metals. Unlike nano- or micro-grained metals, which
strengthen through reduced mobility of dislocations at grain boundaries^{45,46}, nacre’s proteinaceous
organic components contain flexible molecular bonds that elastically accommodate strain and rotation
of nanograins and restoratively return the system to the original state when an external stress is
released.”*

Other comments:

1. On Page 5: “Damping from” should be replaced by “Shearing of” in the opinion of this reviewer.

On page 5, para 3 we have replaced “Damping from” with “shearing of” so it reads *“Shearing of the
interlamellar membranes...”*

2. The authors should comment on the implication of interface shear locking on fracture. The interface
shear deformability and strength appear to be highly dependent on the normal component of the traction.
If the interface is subjected to mainly shear traction, under what conditions would interface locking take
place? Does the experimental data reveal any insights on this?

We have conducted an additional nanoindentation experiment. On page 5 para
3 we describe, “We note, tablets also exhibit a limited amount of locking for
indentation parallel to the tablet plane (Supplemental Fig. 5). This occurred
near the indentation tip where stress is high, and the Poisson effect pushes
the compressed tablet against its neighbors. Further away, unlocked tablets
accommodate shear deformability at their interface and strain contours are
discontinuous. Indentation parallel to the tablet plane was less resilient and
typically resulted in unrecoverable brittle fracture (Supplemental Fig.5).”

Supplementary Figure 5. In situ TEM nanoindentation along a-axis of nacre tablets. (a-d) Compression parallel to the tablet plane showing a limited amount of locking at interface before fracture.

3. On page 6: A) What is the evidence to attribute interface locking a role on the nonlinear elastic
deformation process featuring nanomechanical resilience? In other words, if the tablets were 3-4 times
thicker, would the material loose resilience?

On page 7, para 4 we now state, “When compared to monolithic calcite materials, we clearly see nacre’s
interlamellar membranes reshape compressive strain fields. Both biogenic calcite from the prismatic
layer of *P. nobilis* and geological monolithic aragonite were noticeably stiffer than nacre...”. And as
noted above on page 6, para 2, “Although structurally recoverable locking of tablets is key to nacre’s
resilience, the absorbed energy appears to primarily occur within the resilient deformation of
nanogained tablets that constitute a significant volume fraction (~97%).”

B) The fracture toughness of nacre was first reported by the Espinosa group in the Journal of
Experimental Mechanics, Vol. 47, 2007. The authors may wish to use this reference at the end of the
second paragraph.

We thank the reviewer for this note and have added Barthelat and Espinosa, *Experimental Mechanics*,
Vol. 47, No 3 (2007) to the end of the second paragraph (page 7, para 1).

4. The statement that identifying size effects on mechanical properties of nanostructured materials
necessitates the use of nanoindentation is not fully accurate. A number of other experimental
configurations, e.g., membrane deflection and small-scale tension tests were established to assess size
effects in thin films, nanowires, and other nanomaterials (Espinosa et al., *JMPS*, Vol. 52, 2004;
Ramachandranmoorthy et al, *ACS Nano*, Vol. 9, 2015; Bhowmick et al, *MRS Bulletin*, Vol. 44, 2019).
The authors may wish to revise this.

We appreciate this clarification and have revised the statement on page 8, para 2 so it reads, “*The high*
*strength may be attributed to the finite size of the indentation tip and nanoscale size effects of the*
*mechanical response. As previously observed in several materials—including gold nanowires⁵⁵,*
*polycrystalline thin films⁵⁶, and multiwalled carbon nanotubes⁵⁷—the size effects on mechanical*
*properties of nanostructured materials deviate from bulk and necessitate the use of in-situ*
*nanomechanical testing^{58,59}.”*

5. Could the authors discuss the molecular origin for the “distinct nonlinear elastic deformation
processes that provide high resilience”? Would a nanograined metal or a heavily twinned micrograined
metal exhibit the same resilience? If not, what is different and why it matters?

On page 6 para 3 we now state, “*Here, nacre’s response shows non-linear elastic deformation distinct*
*from that expected in analogous nanocrystalline metals. Unlike nano- or micro-grained metals, which*
*strengthen through reduced mobility of dislocations at grain boundaries^{45,46}, nacre’s proteinaceous*
*organic components contain flexible molecular bonds that elastically accommodate strain and rotation*
*of nanograins and restoratively return the system to the original state when an external stress is*
*released. Energy absorption during protein stretching / unfolding and subsequent energy release upon*
*refolding of the elastomeric molecules provides high resilience in nacre, similar to that found in bone⁴⁷.*
*In contrast, nanocrystalline or nanotwinned metals have lower resilience since they exhibit plasticity*
*via dislocations.”*

**Response to Reviewer 3:**

In their manuscript submitted to Nature Communications, Gim et al. use in situ TEM nanoindentation
measurements on different forms of nacre to provide direct observations of previously postulated
deformation mechanisms in these materials. While most of the mechanisms have been studied before
(and referenced in their paper), I appreciate the systematic approach to this problem by using different
microstructural configurations of nacre and in situ observations. The main conclusions of the paper are
in line with previous studies of these systems, but with some direct observations of the interlocking
behavior and recovery of the organic biopolymer.

We greatly appreciate the positive remarks.

However, the paper doesn’t address what I would consider as significant potential artifacts, and the
calculations of the mechanical data are difficult to interpret. Specifically- the authors seem to calculate
stresses and strains readily in situations that most in the nanomechanics community would hesitate to
do and also seem to ignore the most obvious questions about in situ TEM mechanical testing of organic
beam-sensitive materials.

Here I list what I consider most important problems with the paper:

1) The calculations of strain attenuation (ie- the 3% and the 80% I figure 1) are difficult to interpret.
How meaningful they are in terms of real strain during a mechanical test (is the strain homogeneous
through the sample? Is there a direction or a gradient? Is this true strain or engineering strain? Etc.).
The supplementary figure 5 describes the method, I guess, but there seems to be mixing of elastic and
plastic deformation, so is strain attenuation referring to elastic or plastic strain or both? If it refers to
plastic strain then how meaningful is it- for example, in compression one can simply continue crushing

the sample to increase the ‘strain attenuation’. My point is that I am not sure how meaningful this
measurement is.

To address the ambiguity, we have revised our description to more clearly define our measurements:

Page 5, para 3 now reads “*Strain along the c-axis is highest directly below the tip loading and tablet*
*compression (tablet engineering strain) is measurable using interlamellar demarcation (Supplementary*
*Fig. 6). By ~3% engineering strain in the first tablet, the contours redistribute continuously and by ~6%*
*engineering strain, locking is strikingly visible between tablets. Initially, the engineering strain of the*
*first to second tablet along the axis of indentation decreases by over ~80% when measured using a 0.1*
*um² contact area. This measurement is only one component of an inhomogeneous strain field that on-*
*average, dissipates away from the point of compression. As greater contact stress is applied, the tablets*
*increasingly lock farther away from the tip and the strain attenuation linearly decreases—the*
*deformability is reduced as the nacre behaves more like a monolithic material (Fig. 2c).”*

On page 5, para 3 we also clarify, “*This entire process occurs with elastic processes that are fully*
*repeatable.*” to distinguish the strain attenuation from the less meaningful “crushing” of the sample.

Page 4, para 1 now reads “... However, by 3% engineering strain of the first tablet, ...”.

Page 12, para 1 now reads “The *tablet engineering strain along the c-axis* is defined as the ratio of the
reduction of the tablet width (that is, compressive deformation *directly under the region of loading*) to
its initial width as described in Supplementary Fig. 6. Strain attenuation is defined as the ratio of the
measured *tablet engineering strain* between the first and the second tablet from the indenter tip contact
location (Supplementary Fig. 6).”

Figure 2c now read “*tablet strain attenuation (parallel to c-axis of nacre)*”

2) Furthermore, the calculation of strain leaves some unanswered questions. The indentation stress field
is quite complicated, varying from tensile at the side to compressive underneath the indenter (shear in
between!). So what do these numbers even refer to? I doubt it is possible to accurately quantify stress
in this situation- it is difficult enough when you have indentation onto a flat surface, or linear
compression of a column, but in this case it is a punch on the end of a thin wedge (at best- most of the
figures make it seem that the sample geometries are difficult to define). In fact, the images even show
the varying strain field in the strain contours. This is a major problem with this paper.

We only measure the stress at the point of contact and agree our use of “stress” in the manuscript was
too general. “Stress” has been replaced with “contact stress” throughout the text and figures and figure
captions (Fig 2c, Fig 3a, Fig 4j, Supp. Fig. 5h) when referring to a measurement. Contact stress is
defined in the methods (page 11 para 2): “For all the samples, contact stress is calculated through
dividing the measured load by cross-sectional area of the specimen in contact with the indenter tip.”

As stated above in response to comment 1, we have now made clear that “*Strain along the c-axis is*
*highest directly below the tip loading and tablet compression (tablet engineering strain) is measurable*

*using interlamellar demarcation (Supplementary Fig. 6).*” We agree that quantifying the entire strain
field is challenging and thus has not been reported. Nacre’s unique interlamellar interface structure
provides a unique opportunity to clearly extract compression normal to the tablet surfaces.

As state above in response to comment 1, *“This measurement is only one component of an*
*inhomogeneous strain field that on-average, dissipates away from the point of compression.”*

3) Hydration: in their natural state these materials are hydrated and the mechanical response of the
biopolymer dried, polished and tested in a vacuum must be different. I was surprised to see no mention
of this in the paper.

We have added the statement describing the condition of the specimen in the measurement on page 8,
para 2: *“...the literature reports for bulk nacre in hydrated and dehydrated specimens^{1,5,13-16,27,38,40,43}.*
*Dehydrated nacre has been shown to have a greater strength and elastic modulus, but lower toughness*
*than hydrated nacre due to the plasticizing of the organic matrix by water¹². Here the Pinna nobilis*
*specimen was sacrificed and dehydrated. In native conditions, the performance of nacre should be even*
*better, we underestimate the recoverability of nacre under conditions of low-pressure, low-hydration*
*and overestimate its tendency to fracture.”* On page 10 para 2, we report *“Column pressure in the TEM*
*column at the specimen was $\sim 1 \times 10^7$ torr.”*

4) Electron beam: similar to the hydration argument, the electron irradiation undoubtedly changes the
mechanical properties of the biopolymer (likely as soon as they are first imaged, before any mechanical
tests). Again- the authors don’t address this directly.

On page 10, para 3 we now discuss: *“No change was observed in mechanical behavior measurements*
*with beam exposure: whether the beam was blanked, or the microscope was operated in low-dose STEM*
*or TEM mode. Low-dose methods, beam shuttering, and examination of regions exposed to the beam*
*were used to separate electron beam irradiation from intrinsic phenomena. For STEM measurements,*
*with a typical field of view of 500 nm the electron dose was typically $\sim 150 e^-/\text{\AA}^2$ and dose rates around*
*$\sim 4 e^-/\text{\AA}^2 \cdot s$ the material was structurally preserved during imaging. However, for the same imaging*
*conditions at higher magnifications (e.g. 30 nm field of view) the radiation dose increases to $\sim 2 \cdot 10^5$*
*$e^-/\text{\AA}^2$ and dose rate to $\sim 10^3 e^-/\text{\AA}^2 \cdot s$ which causes the material to show electron irradiation damage*
*localized to the small field of view. Thus, larger fields of view are preferred to minimize dose and*
*provide a large area of observation where fracture may nucleate. For TEM imaging, dose was*
*minimized through use of a heavily diverged beam and a high-efficiency camera (DQE of 0.3) with*
*single electron sensitivity and high-readout speed (up to 300 fps).”*

5) The observations of rotation and deformation of the aragonite nanograins and organic inclusions
(supplementary figure 6) are not at all clear. What exactly is supposed to be rotating in these images
and how do you know they are rotating?

We have added a new Supplemental Movie 4 showing the nanograins deforming under stress.

We have modified Supplementary Figure 7 to highlight the nanograins that are deforming under stress.

We added to the caption, “*Here, BF-TEM contrast of the thin specimen is sensitive to strain and small*
*changes in crystallographic orientation. The visible darkening of grains (green circles) during*
*indentation are due to nanograin deformation or reorientation from local stresses ... The nanogranular*
*response and dynamics are most visible in the Supplemental Movie 4.*”

We have replaced “light up” with “*change contrast*” on page 6 para 2.

We feel that these changes have improved the manuscript and thank the referees for the
helpful suggestions. Please do not hesitate to contact us with any further comments or
requests.

Sincerely yours,

Robert Hovden

Assistant Professor

Materials Science & Engineering

REVIEWERS' COMMENTS:

Reviewer #1 (Remarks to the Author):

The authors have satisfactorily addressed my concerns and questions in their response letter and revised manuscript.

Reviewer #2 (Remarks to the Author):

My comments have been addressed with the exception of one important comment highlighting the differences between nanoindentation and fracture experiments, the later revealing that nacre exhibits a high toughness compared to calcite and aragonite. Indeed, in mode I fracture the process zone induces tension parallel to the tablets. Moreover, the high toughness is obtained only under hydrated conditions!

In the revision the authors state: "Although structurally recoverable locking of tablets is key to nacre's resilience, the absorbed energy appears to primarily occur within the resilient deformation of nanograined tablets that constitute a significant volume fraction (~97%)."

The statement must be revised to indicate that the experiments were performed on dry nacre and under localized compression. Namely, the authors must state:

Although structurally recoverable locking of tablets is key to nacre's resilience, in the reported nanoindentation experiments, performed under dry conditions, the absorbed energy appears to primarily occur within the resilient deformation of nanograined tablets that constitute a significant volume fraction (~97%).

Horacio D. Espinosa

Reviewer #3 (Remarks to the Author):

The authors have adequately responded to my original comments. I recommend accepting this article for publication.

Robert Hovden
Dept. Materials Science
University of Michigan
2300 Hayward St
Ann Arbor, MI 48109
Phone: (770) 265-4042

September 18, 2019

Response to Reviewer 1:

The authors have satisfactorily addressed my concerns and questions in their response letter and revised manuscript.

We thank the reviewer for their support.

Response to Reviewer 2:

My comments have been addressed with the exception of one important comment highlighting the differences between nanoindentation and fracture experiments, the later revealing that nacre exhibits a high toughness compared to calcite and aragonite. Indeed, in mode I fracture the process zone induces tension parallel to the tablets. Moreover, the high toughness is obtained only under hydrated conditions!

In the revision the authors state: “Although structurally recoverable locking of tablets is key to nacre’s resilience, the absorbed energy appears to primarily occur within the resilient deformation of nanograined tablets that constitute a significant volume fraction (~97%).”

The statement must be revised to indicate that the experiments were performed on dry nacre and under localized compression. Namely, the authors must state:

Although structurally recoverable locking of tablets is key to nacre’s resilience, in the reported nanoindentation experiments, performed under dry conditions, the absorbed energy appears to primarily occur within the resilient deformation of nanograined tablets that constitute a significant volume fraction (~97%).

We have revised the sentence accordingly, “Although structurally recoverable locking of tablets is key to nacre’s resilience, *in the reported nanoindentation experiments, performed under dry conditions*, the absorbed energy appears to primarily occur within the resilient deformation of nanograined tablets that constitute a significant volume fraction (~97%).”

We greatly appreciate the careful feedback from Reviewer 2, which has improved the accuracy and interpretation of our manuscript.

Response to Reviewer 3:

The authors have adequately responded to my original comments. I recommend accepting this article for publication.

We thank the reviewer for their support.

We feel that these changes have improved the manuscript and thank the referees for the helpful suggestions. Please do not hesitate to contact us with any further comments or requests.

Sincerely yours,

Robert Hovden
Assistant Professor
Materials Science & Engineering